# Adoption of Transformer Neural Network to Improve the Diagnostic Performance of Oximetry for Obstructive Sleep Apnea

**DOI:** 10.3390/s23187924

**Published:** 2023-09-15

**Authors:** Malak Abdullah Almarshad, Saad Al-Ahmadi, Md Saiful Islam, Ahmed S. BaHammam, Adel Soudani

**Affiliations:** 1Computer Science Department, College of Computer and Information Sciences, King Saud University, Riyadh 11543, Saudi Arabiasaislam@ksu.edu.sa (M.S.I.);; 2Computer Science Department, College of Computer and Information Sciences, Al-Imam Mohammad Ibn Saud Islamic University, Riyadh 11432, Saudi Arabia; 3The University Sleep Disorders Center, Department of Medicine, College of Medicine, King Saud University, Riyadh 11324, Saudi Arabia; 4Strategic Technologies Program of the National Plan for Sciences and Technology and Innovation in the Kingdom of Saudi Arabia, Riyadh 11324, Saudi Arabia

**Keywords:** artificial intelligence (AI), transformer neural networks, deep learning (DL), oxygen saturation (SpO2), polysomnography (PSG), autoscoring, obstructive sleep apnea (OSA)

## Abstract

Scoring polysomnography for obstructive sleep apnea diagnosis is a laborious, long, and costly process. Machine learning approaches, such as deep neural networks, can reduce scoring time and costs. However, most methods require prior filtering and preprocessing of the raw signal. Our work presents a novel method for diagnosing obstructive sleep apnea using a transformer neural network with learnable positional encoding, which outperforms existing state-of-the-art solutions. This approach has the potential to improve the diagnostic performance of oximetry for obstructive sleep apnea and reduce the time and costs associated with traditional polysomnography. Contrary to existing approaches, our approach performs annotations at one-second granularity. Allowing physicians to interpret the model’s outcome. In addition, we tested different positional encoding designs as the first layer of the model, and the best results were achieved using a learnable positional encoding based on an autoencoder with structural novelty. In addition, we tried different temporal resolutions with various granularity levels from 1 to 360 s. All experiments were carried out on an independent test set from the public OSASUD dataset and showed that our approach outperforms current state-of-the-art solutions with a satisfactory AUC of 0.89, accuracy of 0.80, and F1-score of 0.79.

## 1. Introduction

Obstructive sleep apnea (OSA) is a sleep disorder that occurs when the upper airway collapses during sleep, blocking airflow [1]. This can lead to repeated episodes of shallow or interrupted breathing. OSA is a common medical condition affecting an estimated one billion people worldwide [2]. The prevalence of OSA is particularly high in middle-aged and older adults, with some studies reporting a prevalence of up to 50% in these age groups [2]. Untreated OSA can have serious health consequences, including heart disease, stroke, diabetes, and impaired quality of life [2]. One reason for this is the delayed diagnosis and treatment due to the complex diagnostic procedures required to conduct and interpret sleep studies [3].

Polysomnography (PSG) is considered the gold-standard diagnostic test for OSA, where the patient undergoes neuro-muscular-cardio-respiratory monitoring in the sleep laboratory [4]. PSG can be a full-night diagnostic study followed by a full-night therapeutic study or split-night sleep study, where the first half is diagnostic and the second half is therapeutic. The American Academy of Sleep Medicine (AASM) guidelines for PSG performance recommend collecting the following signals: electroencephalogram (EEG), electrocardiogram (ECG), electromyography (EMG) for chin and legs, thermal sensors, nasal pressure-flow transducer, and photoplethysmography (PPG) [5,6]. PSG is expressed as epochs of raw data, where each epoch is 30 s (Figure 1). A record of 8 h translates to around 900 pages (epochs) [4].

Even though PSG is the ideal method for OSA diagnosis, there are some inherent constraints and shortcomings in PSG; for instance, a type-I sleep study requires admission to the sleep laboratory and a sleep technician attendance, and subsequently, a well-trained technician spends a lot of time scoring PSG manually from the start to the end. Moreover, each PSG record might be scored by multiple technicians. Sleep medicine physicians frequently rely on the manually scored reports provided by technicians to make their medical decisions, and this is compounded by the significant backlog of patients waiting for medical attention. Furthermore, readings given by sleep technicians for OSA events are also subject to inter-scorer variability among technologists [7]. These differences are likely due to diverse rules used to score events as well as differences in the technologist’s understanding of the rules [7]. The whole process is complicated and expensive, which could lead to delayed diagnosis and treatment for patients with OSA [7]. Considering the high prevalence and the serious consequences of OSA, much more effort is needed for accurate and early diagnosis [2].

**Figure 1 sensors-23-07924-f001:**
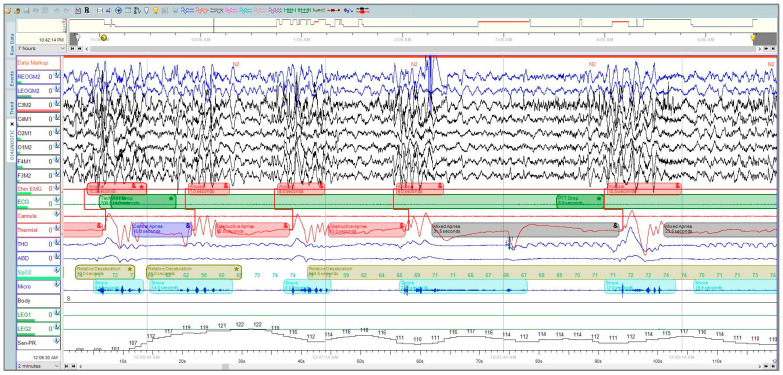
A polysomnographic recording from Sleepware G3 [8] shows a two-minute window that was manually scored by a professional technician at the University Sleep Disorders Center at King Saud University Medical City (KSUMC).

Blood oxygen saturation (SpO2) stands for the saturation percentage of oxygen in hemoglobin. It is a measure of how much oxygen is bound to hemoglobin in the blood. SpO2 is typically measured using a pulse oximeter, which is a small, handheld device that clips onto a finger or earlobe [5]. The AASM recommends a fast sampling rate oximeter (shorter interval, e.g., 3 s or less) to improve sensitivity as patients with sleep-disordered breathing (SDB) usually have short-lasting intermittent hypoxemia, which can be missed if the sampling rate is too slow [9]. Smartwatches monitor SpO2 with a high level of accuracy [5,10,11]. Recently, certain advanced smartwatches and fitness trackers have been equipped with sleep-tracking features that can detect and indicate the potential presence of sleep apnea [10]. Patients suffering from OSA have intermittent oxygen desaturation that follows obstructive respiratory events. Desaturation during sleep studies is scored when there is a drop of SpO2 of 3% or 4% (depending on the used criteria) from the previous normal SpO2 [6]. Generally, the aim is always to keep oxygen saturation levels above 88%. All of this suggests that SpO2 monitoring using pulse oximetry or other wearable technology could be an effective and affordable tool for early OSA screening [5,12]. In addition, it encourages the development of machine learning (ML) and deep learning (DL) models that detect OSA utilizing SpO2 exclusively [13]. The use of SpO2 only for OSA screening has several benefits [14] Pulse oximetry is non-invasive, cost-effective, portable and convenient. This means that it does not require any electrodes or sensors to be placed on the body, and it does not cost as much as other methods, and can be used in home settings [15]. Theoretical research on the use of SpO2 only to automate OSA scoring has been limited. However, further investigation is needed to confirm these findings and determine the validity of using DL to classify OSA from SpO2.

Over the last 10 years, few PSG auto-scoring solutions have evolved. Among them, the only ready-to-use product is Somnolyzer from Philips [8]. Somnolyzer is proprietary software, which means that the algorithm behind it is not publicly available. This makes it difficult to assess the accuracy and reliability of the software [16]. Theoretically, multiple types of research have been conducted to automate OSA scoring. These include statistical analysis, signal processing, ML [13], and DL methods. A study in 2019 suggested that the DL approach for sleep event detection can reach expert human performance [17]. Furthermore, features in DL are learned during the training and not handcrafted by a human; this is a big advantage over other ML algorithms [18,19]. Deep learning models are trained by feeding them a large amount of data and adjusting the weights of the connections between the nodes until the model can make accurate predictions. The training process is typically performed using the backpropagation algorithm, which gradually adjusts the weights of the connections to minimize the error between the model’s predictions and the ground truth labels. Once a deep learning model is trained, it can be used to make predictions on new data [14,19].

This paper presents a transformer-based DL framework with a novel learnable positional encoding for the effective classification of raw SpO2 as time series data into normal and apnea segments. Positional encoding is a powerful technique that can help transformer models to better understand the order of a sample in a segment. Positional encoding is added to the segment embeddings before they are passed to the transformer model. This allows the model to learn the relationship between the position of a sample in a segment and its surroundings. Such an architecture can be used as a part of a larger system to preprocess raw signals before further elaboration. Unlike previous DL solutions applied to OSA detection, the proposed architecture is specially designed to handle raw SpO2 signals with an arbitrary noise, keeping temporal relationships over long time windows. Moreover, OSA events are labeled at fine granularity. Such an ability provides physicians with detailed information about the condition of the patient, allowing them to understand the results of the model and accelerate the diagnosis process.

## 2. Related Work

Recent research has shown that DL can be used to detect sleep apnea using SpO2 only with an acceptable degree of accuracy [20]. For example, one study found that a DL model based on an ANN architecture was able to detect sleep apnea with an accuracy of 97.8% [21]. Generally, four main types of DL networks were widely used to detect OSA from SpO2 [20]. Earlier, a deep vanilla neural network (DNN) was used to learn simple patterns [22,23]. More recent work tends to use a convolutional neural network (CNN) to learn spatial features from sleep apnea data [12,24,25,26,27]. Lately, temporal patterns have been learned using recurrent neural networks (RNNs), long short-term memory (LSTM), or a hybrid architecture [28,29]. While few researchers prefer to build their own datasets from scratch, many prefer benchmark datasets. Two datasets have been used in the literature, namely St. Vincent’s University Hospital/University College Dublin Sleep Apnea (UCD database) [30] and the Childhood Adenotonsillectomy Trial (CHAT) dataset [31]. Some researchers have used only SpO2 to diagnose apnea using DL, while others have used other signals combined with SpO2, such as ECG, EEG, and respiratory effort, for more accurate diagnosis [21,27,28]. Previous attempts to automate the diagnosis of OSA from SpO2 have many shortcomings. Intensive filtering methods are applied to reduce the noise [28,29]. Furthermore, some of them heavily rely on data preprocessing and feature extraction [23,27]. Cen et al. [32] used only accuracy to measure the performance of an imbalanced dataset, which is not always a reliable metric. Table 1 provides a brief chronological order list of the approaches that make use of oximetry to detect OSA using DL.

## 3. Materials and Methods

We used the OSASUD dataset in this study [34]. In the first set of experiments, we tried to determine the best positional encoding mechanism. In the second set of experiments, we tried training the best model with different SpO2 sequence lengths, starting from 10 s to 360 s. Figure 2 shows the schematic diagram of the generic part of the proposed architecture. We refer the reader to the original Transformer paper for a detailed description of the model [35] and here explain the proposed modifications that make it suit continuous univariate time series classification instead of generating sequences of discrete tokens. The input data are extracted from the OSASUD public dataset. The raw signal is then normalized and fed to the model as batches of 32 samples with different sequence lengths from 1 s to 360 s. The proposed convolutional autoencoder (CAE) then learns the best representation for each batch to pass it as an input to a stack of four layers of the Transformer encoder.

All experiments have been executed on a local machine equipped with AMD Ryzen™ 9 5900X series CPU, NVIDIA GeForce RTX 3080 GPU [36], and 32 GB of RAM. As for the development framework, we relied on TensorFlow 2.10 [37]. For reproducibility purposes, the source code of the developed model is available online at https://github.com/malakalmarshad/Transformer_SpO2 (accessed on 13 September 2023).

### 3.1. Dataset

We retrieved the data from the Obstructive Sleep Apnea Stroke Unit Dataset (OSASUD), which is publicly available on Figshare [34]. OSASUD contains overnight vital signs data from multi-channel ECG, photoplethysmography, and other class III device (cardio-respiratory monitoring) channels, combined with related domain experts’ OSA annotations for 30 patients. The full database version was downloaded. The raw dataset contained around 1 million data records.

Deep learning (DL) models are able to extract features from data without the need for extensive preprocessing. Therefore, we chose not to apply any filters to the OSUSA dataset, which is known to contain high levels of noise. Our DL model is capable of dealing with this noise and still achieving good performance [34]. All input arrays were independently rescaled and mapped between 0 and 1, using min-max normalization [19]. However, we did not find a noticeable difference in the model’s outcome if Z-Normalization (Standardization) is used. Transforming data into one uniform scale helps stabilize the gradient descent optimization algorithm, the basic algorithm to train DL models [38]. Moreover, 132,580 NaN values are dropped [39].

### 3.2. Base-Model

The core of our approach to classifying apnea events is a Transformer encoder, which was first described in the paper “Attention Is All You Need” by Vaswani et al. (2017) [35]. The encoder maps a raw SpO2 sequence X = [$x_{0}$, …, $x_{k}$,…, $x_{n-1}$, $x_{n}$ ], to an apnea or normal sequence. The Transformer is established on an attentional mechanism only. It lacks any sort of convolutional or recurrent components to preserve long sequence dependencies. Transformer uses self-attention to compute the representations of input and output sequences. Self-attention refers to the ability of a model to attend to different parts of the same sequence when making predictions. We stacked four identical layers of the encoder component. Each layer of the transformer encoder is composed of a multi-head self-attention and feedforward sublayer. The attention layer at first calculates three vectors from each ‘sample’ of a ‘sequence’, key, query, and value. To process all samples in a sequence simultaneously, key vectors are combined together into matrix *K*, and queries and values produce matrices *Q* and *V* correspondingly. The attention is computed as follows:(1)$$\mathrm{Attention}(Q,\;K,\;V)=\mathrm{softmax}(\frac{QK^{T}}{\sqrt{d_{k}}})V$$
where *d* is the dimension of the hidden states, and the hidden states are the representations of the input sequence that are used by the attention mechanism to compute the attention weights. The dimension of the hidden states is typically set to be the same as the number of neurons in the attention layer. The Transformer architecture outperforms its predecessors due to the multi-head attention mechanism and the positional embedding components. Unlike RNNs and CNNs, Transformers do not contain recurrence or convolution, which allows them to preserve information about the relative or absolute position of the samples. To achieve this, Transformers add a positional encoding component to the input embedding at the bottom of the encoder and decoder stacks. This information is then echoed until the last stack of transformer blocks [35].

The model takes a tensor of shape (batch size, sequence length, features) as an input, where the batch size is the number of samples that will be used to train the model at once, sequence length is the number of time steps passed each time, and features are the normalized raw SpO2 values.

The order of the sequence is an essential part of time series data. CNNs, RNNs, and LSTMs inherently take the order of the sequence into account, while Transformers ditch the recurrence mechanism in favor of a multi-head self-attention mechanism to parallelize the training and make it faster. We experimented with three different positional embedding strategies to preserve the order of the sequence in Transformers: naive constant position embedding (i.e., index of the sample), relative positioning (sinusoidal positional embedding), and a novel learned positional embedding [40,41,42].

Without positional encoding, samples are treated like a bag of words. Sequence positional embedding is directly added with the sequence representation, as the following:(2)$$Z_{i}=\mathrm{inputE}\;(x_{i})+PE\;{}_{\left(i\right)}$$ where $x_{i}$ is the sequence at the *i*-*th* position, *inputE* is the input embedding, and *PE* is the positional encoding, which can be either a learnable embedding or a predefined function.

A flowchart diagram of the detailed phases of the conducted deep learning (DL) model is shown in Figure 3. The chart begins with data extraction, followed by data preprocessing, then model training, and lastly, model evaluation using a test set that is never seen by the model.

#### 3.2.1. Constant Position Embeddings

It is a finite-dimensional representation of the sample index in a sequence. Given a sequence, *X* = [$x_{0}$, …, $x_{k}$, $x_{n-1}$, $x_{n}$ ], positional encoding is a tensor that is fed to a model to tell it where some value $x_{i}$ is in the sequence *X*. Fixed positional encoding based on the normalized index of the sequence, simply calculated as the following:(3)$$PE\;(i)=\frac{pos(x_{i})-min(pos(x))}{max(pos(x))-min(pos(x))}$$
where *pos* is the position and *i* is the dimension.

#### 3.2.2. Sinusoidal Positional Embedding

Sinusoidal positional encoding works by creating a vector for each position in a sequence. The vector is a combination of *sine* and *cosine* functions, with the frequency of the functions depending on the position. It allows the model to learn the relative position of a sample in a sequence. Using sine and cosine functions of different frequencies.

The sinusoidal positional encoding encodes the position along the sequence into a vector of size *^d^_model_*, as described in the transformer paper [35]:*PE*_(*pos*,2*i*)_ = *sin* (*pos*/10,000^2*i*/*d*^*_model_*)
*PE*
_(*pos*,2*i*+1)_ = *cos* (*pos*/10,000^2*i*/*d*^*_model_*)
(4)
where *pos* is the position, and *i* is the dimension. 2*i* and 2*i* + 1 are used to alternate between even and odd sequences. We tried different lengths and depths of the sinusoidal embedding. Setting the length to 64 and depth (*^d^_model_*) to 32 appeared more reasonable for the dataset.

#### 3.2.3. Learned Positional Embedding

Here, we replace the traditional absolute positional encoding component with a simple convolutional autoencoder (CAE). It consists of a stack of convolutional layers and transposed convolutional layers [43], with dropout layers in between to prevent overfitting. Overfitting occurs when the model learns the training data too well and is unable to generalize to new data [18]. The dropout layers are a generalization mechanism that randomly drops out some of the neurons in the model during training, which helps to prevent the model from memorizing the training data [19]. This part of the model is the first component, and it takes a 1D input sequence and outputs a reconstruction of the same sequence.

Autoencoders are a type of nonlinear dimensionality reduction that is more powerful than principal component analysis (PCA) [19]. PCA is a linear dimensionality reduction method. Both are examples of self-supervised ML techniques, where the generated target is an approximation of the input. Autoencoders consist of two components: (i) An encoder: This component maps the input data to a latent space of lower dimensionality. (ii) A decoder: This component maps the latent space back to the original input space. The encoder learns to identify the most important features of the input data, while the decoder learns to reconstruct the input data from the latent space [44]. This allows autoencoders to learn more complex representations of the data than PCA [18]. Autoencoders are useful for data denoising, dimensionality reduction, and learning a better representation of the samples’ distribution [19]. The key task was to fine-tune an autoencoder to fit the job by determining the number of layers, the different filters in those layers, and the kernel size. The kernel size defines the size of the sliding window [45].

In this work, the first few layers of the CAE are convolutional layers with a kernel size of seven and a stride of two. This means that the model will learn to extract features from the input sequence that are seven timesteps long. The activation function for these layers is ReLU, which is a nonlinear function that helps the model learn more complex features. The next few layers are transposed convolutional layers [43]. These layers are used to reconstruct the input sequence from the features that were extracted by the convolutional layers [46]. The activation function for these layers is also ReLU. The final layer of the CAE is a convolutional layer with a kernel size of seven and a stride of one. This layer outputs a 1D sequence of the same length as the input sequence. Then, the reconstructed sequence is fed to the encoder part of the Transformer. We chose convolutional autoencoders over feedforward autoencoders because convolutional layers are better at capturing spatial information. Spatial information is the arrangement of features in a data set, such as the location of pixels in an image or the order of words in a sentence. Convolutional layers can learn to identify patterns in spatial data, allowing them to reconstruct the data more accurately [47].

### 3.3. Determining the Best Sequence Length

Sequence length is an important consideration for time series analysis, as it can affect the accuracy of the results. For example, when attempting to predict future values within a time series, it is crucial to ensure that the sequence length is sufficient to capture the underlying patterns present in the data. However, if the sequence length is too long, it can make the analysis more computationally expensive. It is best to start determining the optimal sequence length using domain knowledge. For example, if we are trying to predict OSA from the SpO2 time series, we know that the patterns in the PSG data are typically scored by sleep technologists using a two-minute time window [4]. In this case, you would want to choose a sequence length that is around this value. We tried varying segment length from 10 s to 6 min, and the performance of the model varied accordingly. Detailed performance metrics for each segment are provided in the results section (Section 4).

### 3.4. Experimental Setting

Following Occam’s razor, we started with the simplest model and added layers of increasing complexity only as needed. We performed a series of experiments to evaluate the performance of each model on a real-world dataset of around 1 million samples. This process allowed us to develop a model that was both accurate and efficient. To tune the hyperparameters of our model, we used a random search using K-fold cross-validation. K-fold cross-validation is a statistical method for evaluating ML models [19]. In K-fold cross-validation, the dataset is randomly divided into K folds of equal size. One of the folds is used as a validation set, and the other *K*-1 folds are used for training. This process is repeated *K* times, and the results are averaged to obtain an estimate of the model’s performance. K-fold cross-validation is less prone to variation than other methods because it uses the entire training set and tunes the model against a subset of itself. In our experiment, we used *K* = 5-fold cross-validation. From the beginning, the dataset was divided into two parts: 20% was kept aside for final testing and 80% for training. The training dataset was then divided again into five folds, with 20% of the data in each fold used for validation.

We performed two sets of experiments. In the first set, we evaluated three different positional encodings: sinusoidal, constant, and learnable. In the second set, we experimented with different sequence lengths, from 10 to 360. By performing these experiments, you would be able to determine which positional encoding and sequence length works best for OSA detection.

### 3.5. Evaluation Metrics

True positive (TP) are samples correctly identified as apnea, and true negative (TN) are the samples correctly identified as normal. The considered evaluation metrics that are calculated from the confusion matrix (Table 2) include accuracy, recall (sensitivity), precision, f1-score, and area under the receiver operating characteristic curve (ROC AUC).

*Accuracy*: measures how often the model is making a correct prediction.

*Sensitivity or (recall)*: how capable can the model identify apnea events?

*Precision*: out of all the samples that were predicted as positive, how many are correctly positive?

*F*1-*Score*: a weighted average of recall and precision.

The four measures are given by Equations (5)–(8):(5)$$Accuracy=\frac{TP+TN}{TP+TN+FN+FP}$$
(6)$$Recall=\frac{TP}{TP+FN^{\prime }}$$
(7)$$Precision=\frac{TP}{TP+FP}$$
(8)$$F1-Score=\frac{2\times Precision\times Recall}{Precision+Recall}$$

Since the used dataset is imbalanced, accuracy alone is not sufficient to correctly evaluate the model’s performance. This is because there are fewer anomalies than normal events in the dataset, which means that the model can achieve a high accuracy simply by predicting all events as normal. To overcome this limitation, we also took into account other metrics. In general, it is good practice to track multiple metrics when developing a DL model, as each highlights distinct aspects of model performance. In addition, we considered the area under the receiver operating characteristic curve (ROC), which plots the true positive rate (TPR) against the false positive rate (FPR). This curve illustrates how well the model can differentiate between the two classes, where a random model cannot exceed 0.5.

## 4. Results

Initially, we ran four experiments by slightly modifying the model each time. All of these experiments were conducted on the OSASUD dataset. Results are shown in Table 3. Starting with only an encoder part of the Transformer without any positional embedding, we applied Adam as an optimizer [48]. In the second and third experiments, we added the order of the samples for each batch as described in the naive positional encoding and sinusoidal positional encoding sections, respectively. We observed that these two types of constant positional encoding did not significantly improve the model’s overall performance. In the fourth experiment, the static positional encoding part was removed, and a learnable positional encoding employing a convolutional autoencoder (CAE) was added to the model just before the Transformer encoder. Table 3 shows that this structure achieved the best performance across AUC, accuracy, and sensitivity.

The batch size in these four experiments was set to 32, and the sequence length was 180. The initial learning rate was set to 1 × 10^−5^, which is a very small learning rate. This will prevent the model from diverging, but it is still large enough to allow the model to learn effectively. If the validation loss curve starts to plateau, the learning rate will be reduced by a factor of 0.2.

In the second set of experiments, we trained the best model with SpO2 sequence lengths ranging from 10 s to 360 s. We wanted to determine the optimal sequence length for the model to achieve the best performance. As seen in Figure 4, both the AUC and accuracy increase as the sequence length increases. In addition, Table 4 shows that longer sequences require more computation resources and take longer time to converge.

By applying the same performance metrics as Bernardini et al. [28], Table 5 shows that our models considerably outperform previous proposals, with the overall best results provided using the Transformer-based classification, considering per segment results. More precisely, for per second results, an AUC, accuracy, and F1 improvement of 2.04%, 1.45%, and 3.7%, respectively, was obtained compared with state-of-the-art (SOTA) approaches. In Table 5, the average, minimum, maximum and standard deviation are aggregated from Table A1 in the Appendix A.

## 5. Discussion

We can observe from the beginning that Transformer-based models are able to learn positional information even when they are not provided with any explicit positional encoding [49]. In addition, Transformer-based models with learnable positional encoding using a CAE yielded a better performance for OSA classification compared to previous SOTA architectures (Table 3 and Table 5). The addition of the CAE enables the model to capture the temporal dependency. It also projects the input data, epoch by epoch, into an enhanced representation that improves the awareness of the sample self-position on each time step and its surroundings. The superiority of CAEs over other positional encodings was not surprising, given that convolutional autoencoders have been shown to be effective for time series encoding tasks in the past, such as the Rocket [43,45].

Considering all metrics, the best sequence length for this model is 120 s (Figure 4). This suggests that the model is able to learn sufficiently about the SpO2 signal when given enough data. 120 s is relatively short compared to the 180 s used by Bernardini et al. [28] or even other segment lengths used in our experiments. Using short-length time series segments in neural networks is influential for a few reasons. First, it can help to improve the accuracy of the model; short-length segments are less likely to be affected by noise or outliers, which can degrade the performance of the model. Second, short-length segments can make the model more efficient because they require less data to train the model, which can lead to faster training times and lower computational costs. Finally, short-length segments can make the model more interpretable and easier to understand than long sequences, which can help to identify the patterns that the model is learning [15].

Additionally, we encountered memory errors when we tried to train the first three architectures (Transformer encoder alone, constant positional encoding, and Sinusoidal positional encoding) with a sequence length of more than 180. The machine we were using could not handle the large amount of data required for these models. On the contrary, the use of the autoencoder as positional encoding allowed us to expand the sequence length to 360 using the same machine. This means that the model utilizes the memory more efficiently and is able to digest more data at once.

Existing approaches for OSA classification typically use annotations at a coarser granularity, such as 30-s or 60-s intervals (Table 1). This can make it difficult for physicians to interpret the model’s outcome, as they may not be able to see the subtle changes in the sleep signal that are associated with different sleep events. Our approach, on the other hand, performs annotations at a one-second granularity. This allows physicians to see the sleep signal in much more detail, which can help them to better understand the model’s outcome and to make more accurate diagnoses. Compared to SOTA with different datasets, our model achieved 0.908 for AUC (Table 5). This indicates that the proposed classifier can distinguish between positive and negative examples very well. Nevertheless, to make a fair comparison, the models should be trained and evaluated on the same dataset. However, even though Bernardini et al. used the same dataset [28], our model was tested on an independent test split that was kept aside before the training. AIOSA [28] results were produced by splitting the dataset into two sets, one for training and the other for validation, which might lead to data leakage and raise questions about the generality of the obtained results [19]. Figure 5 shows the variable accuracy, loss, learning rate, and ROC curve of the proposed model for the first 40 epochs.

As for per-patient detection, our model provides perfect classification accuracy. This is quite interesting, as it supports the intuition that the proposed Transformer-based architecture can effectively summarize input data while preserving its temporal content. The last rows of Table 5 focus on the per-patient results obtained using the proposed model compared to AIOSA [28] results. The proposed model scored high accuracy, ROU AUC, and low F1 scores, especially in patients with higher AHI (severe OSA). In the context of OSA detection, it is important to prioritize accuracy over the F1 score because the cost of false negatives is high. A false negative (predicting that a patient does not have OSA when they do) can lead to OSA going undetected, which can have serious health consequences. As a final remark, we noticed that, although F1 scores may seem low for some patients, they are explainable using the corresponding very high sensitivity and accuracy values. This is the case, for instance, with patient 10 and patient 21, whose AHI is very high. The detailed results of each patient as a test set are presented in Appendix A (Table A1, p. 14. Figure 6 shows predicted and ground truth anomalies for patients 19, 5, and 14, with different OSA severity.

Our work differs from existing ones in at least four fundamental aspects. First, we used a real-life dataset that is both noisy and highly skewed. Second, we detected apnea on a 1-s granularity. Third, our model is able to digest raw signals with a high sampling frequency, maintaining temporal and spatial relationships over large time windows. Fourth, the model was tested against an entirely new raw set of data that was kept aside during the model training phase.

The proposed work suffers from potential limitations. First, this study only used the OSASUD dataset, which was collected from a single hospital. Although the amount of data in this database is sufficient, we still think that additional experiments on another dataset could be useful. In addition, although models based on SpO2 only to detect OSA are common in the DL literature, sleep apnea is directly related to respiration, and AASM only approved a minimum of 3 channels for HSAT [4]. Therefore, the reliability of only SpO2 for OSA detection needs further investigation to be clinically approved. Furthermore, the use of more than one signal from multiple sensors might improve the model’s predictive capability [28]. Moreover, some of this work’s limitations are inherited from the OSASUD dataset [34]. The dataset was extracted from a class III device that lacks EEG channels. For instance, periods of wakefulness after sleep onset and hypopneas associated with arousals but without significant desaturation may have been overlooked [34].

In the future, we plan to test the framework on different datasets and use different PSG signals. This will help to further evaluate the performance of the framework and to determine its clinical utility. We also plan to explore the use of the framework to distinguish between different sleep apnea disorders. This study is conducted using neural networks to solve a decision problem. However, there are other advanced optimization algorithms that can also be effective for challenging decision problems [50,51,52]. In future research, we will explore these algorithms and compare their performance with deep learning.

## 6. Conclusions

This paper presented a DL framework for OSA detection based on transformers using only SpO2. The framework is based on a transformer encoder with a convolutional autoencoder as a positional encoding. This approach outperformed other positional encodings, such as naïve and relative positional encodings. The proposed model was validated on a public dataset collected from real-life scenarios and showed superior performance over existing solutions. Furthermore, the model was further evaluated on a different set of patients that were completely unseen during training, which confirmed its reliability. It achieved an accuracy score of 0.82 and an AUC-ROC of 0.90. These results are comparable to the SOTA performance.

The framework is a practical and efficient tool that has the potential to improve the clinical management of OSA and reduce the burden of this disease. The framework can be used to detect OSA from a noisy waveform without prior intensive preprocessing. This makes it a valuable tool for clinicians who are looking for a quick and easy way to screen for OSA. Overall, the proposed work is a promising step towards the development of a non-invasive, wearable device for OSA detection. However, the results need to be validated on a larger dataset and with other sensors before they can be clinically approved.

## Figures and Tables

**Figure 2 sensors-23-07924-f002:**
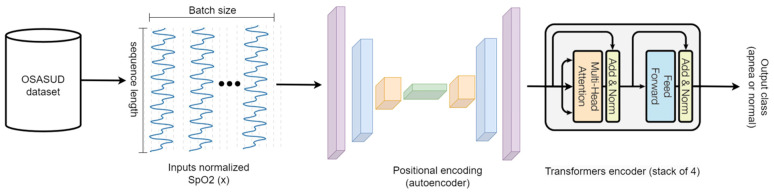
Proposed Transformers-based model general architecture.

**Figure 3 sensors-23-07924-f003:**
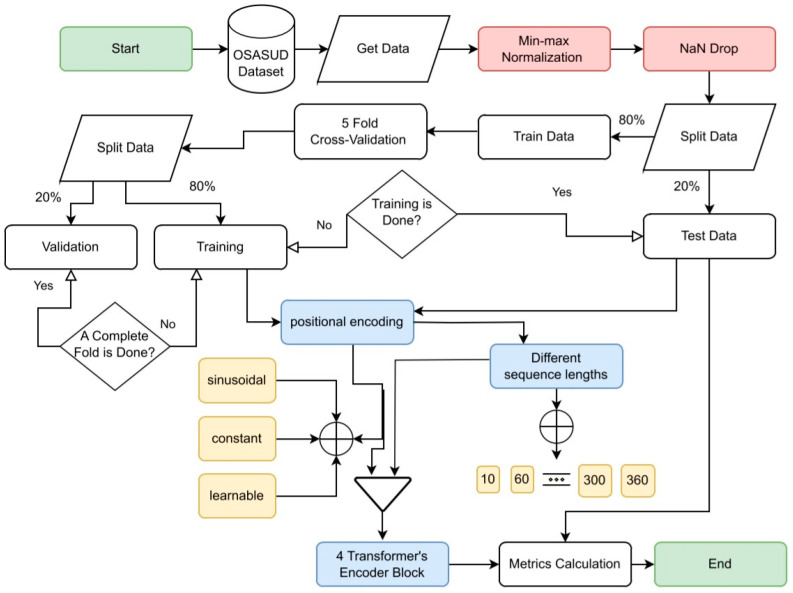
A flowchart describing the general workflow for a deep learning (DL) model. Green rounded rectangles represent the start and the end of the DL model. The preprocessing phase is in red, and the base-model steps are in blue, while different experiments are in yellow. The rest are the usual phases of a deep learning model.

**Figure 4 sensors-23-07924-f004:**
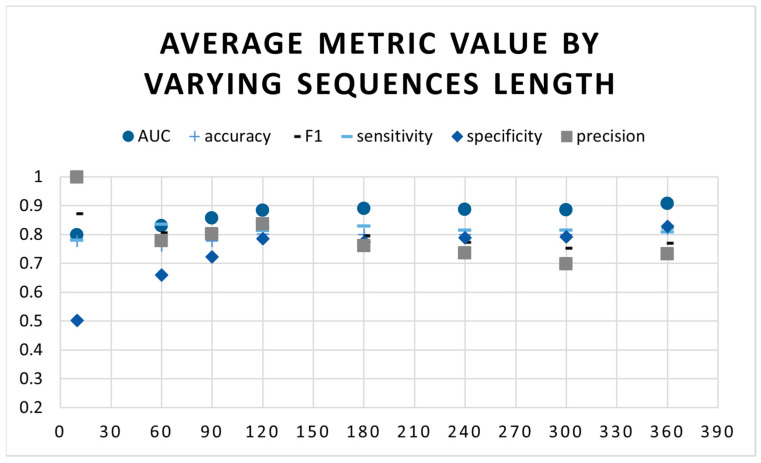
Model results using learnable positional encoding by varying the sequence length.

**Figure 5 sensors-23-07924-f005:**
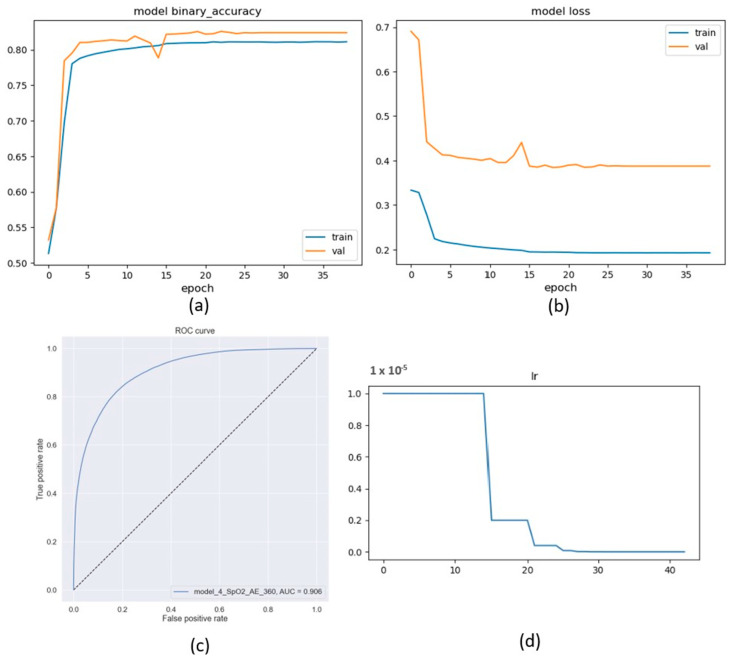
Transformer-based architecture after adding the autoencoder component: (**a**) training and validation accuracy. (**b**) training and validation loss, (**c**) ROC curve, and (**d**) Learning rate decreased after the 10th epoch.

**Figure 6 sensors-23-07924-f006:**
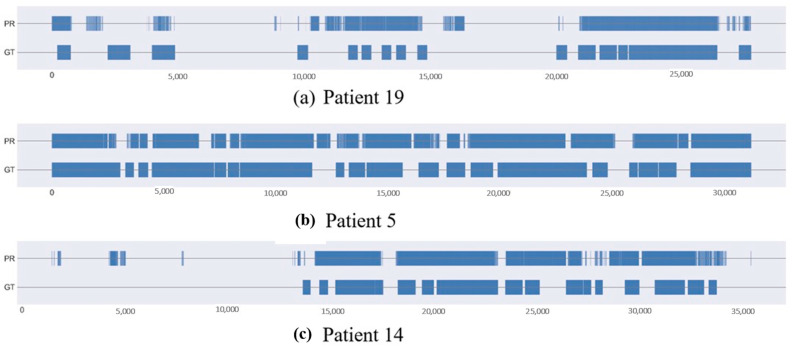
Predicted (PR) and ground truth (GT) events plot for three patients.

**Table 1 sensors-23-07924-t001:** Different approaches for apnea detection from SpO2 using DL, arranged by year of publication.

Ref	Year	DL Model	Dataset	Window Size (Time)	#* Subjects	Accuracy % (Best)
Almazaydeh et al. [23]	2012	NN *	UCD database [30]	-	7	93.3
Morillo et al. [22]	2013	PNN *	Private dataset	30 s	115	84
Mostafa et al. [26]	2017	Deep Belief NN with an autoencoder	UCD database [30]	1 min	8 and 25	85.26
Pathinarupothi et al. [29]	2017	LSTM *-RNN	UCD database [30]	1 min	35	95.5
Cen et al. [32]	2018	CNN *	UCD database [30]	1 s	-	79.61
Mostafa et al. [33]	2020	CNN	Private dataset and UCD database [30]	1, 3 and 5 min	-	89.40
John et al. [12]	2021	1D CNN	UCD database [30]	1 s	25	89.75
Vaquerizo-Villar et al. [25]	2021	CNN	CHAT dataset [31] and 2 private datasets	20 min	3196	83.9
Piorecky et al. [27]	2021	CNN	Private dataset	10 s	175	84
Bernardini et al. [28]	2021	LSTM	OSASUD [34]	180 s	30	63.3
Li et al. [21]	2021	Artificial neural network (ANN)	Private dataset	-	148	97.8

* Abbreviations: #, number of; NN, neural network; PNN, Probabilistic neural network; UCD, St. Vincent’s University Hospital/University College Dublin Sleep Apnea; LSTM, long short-term memory; CNN, convolutional neural network.

**Table 2 sensors-23-07924-t002:** Confusion matrix.

	Ground Truth (*GT*)	
Predicted (PR)	True Positive (*TP*)	False Positive (*FP*)
	False Negative (*FN*)	True Negative (*TN*)

**Table 3 sensors-23-07924-t003:** Performance comparison on the OSASUD dataset using different positional encodings of the Transformer architectures.

Architecture	AUC	Accuracy	F1	Sensitivity	Specificity	Precision
Transformer encoder only	0.8738	0.7898	0.8092	0.7473	0.8526	0.8822
Transformer encoder with naïve position embeddings	0.8788	0.7969	0.8105	0.7665	0.8366	0.8598
Transformer encoder with sinusoidal positional embedding	0.8799	0.7983	0.8118	0.7678	0.8381	0.8610
Transformer encoder with Learned Positional Embedding	0.8890	0.7995	0.7931	0.8285	0.7745	0.7605

**Table 4 sensors-23-07924-t004:** Model performance summary for different Sequence lengths.

Sequence Length	Duration	Trainable Parameter
360	16 h 15 min	139,884
300	6 h 33 min	132,204
240	5 h 17 min	124,524
180	4 h 56 min	116,844
120	4 h 3 min	109,164
90	2 h 45 min	105,580
60	3 h 55 min	101,484
10	1 h 12 min	95,340

**Table 5 sensors-23-07924-t005:** Performance comparison on the OSASUD dataset considering per second and per-patient.

			Per Second					
Model	Base Architecture	Dataset	AUC	Acc *	F1	Sens *	Spec *	Prec *
Mostafa et al. [33]	CNN	Private	-	89.40	-	74.40	93.90	-
		UCD	-	66.79	-	85.37	60.94	-
Morillo et al. [22]	PNN	Private	0.889	85.22	-	92.4	95.9	-
Bernardini et al. [28]	LSTM	OSAUCD	0.704	0.676	0.399	0.656	0.680	-
proposed	Transformer		0.908	0.821	0.769	0.808	0.829	0.733
			Per-patient (OSA = AHI ≥ 5)					
Bernardini et al. [28]	LSTM	OSAUCD	-	0.633	0.776	0.826	0.0	-
Proposed (AVG *)	Transformer		0.868	0.804	0.422	0.647	0.801	0.385
Proposed (MAX *)	Transformer	OSAUCD	0.990	0.968	0.826	0.996	0.992	0.776
Proposed (MIN *)			0.553	0.522	0.034	0.061	0.235	0.0176
Proposed (σ *)			0.119	0.121	0.254	0.295	0.162	0.249

* Abbreviations: Acc, accuracy; Sens, sensitivity; Spec, specificity; Prec, precision; AVG, Average; MAX, maximum; MIN, minimum; σ, Standard deviation.

## Data Availability

The authors confirm that the data supporting the findings of this study are available within the article.

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
