# Peer review of "Adoption of Transformer Neural Network to Improve the Diagnostic Performance of Oximetry for Obstructive Sleep Apnea"

_sensors, 2023, doi:10.3390/s23187924_

Round 1
Reviewer 1 Report
The authors have proposed a Transformer-based network for obstructive sleep apnea. The manuscript is complete, and the authors try to prove the progressiveness of the algorithm through experiments. However, there are some problems that need to be revised. The comments are as follows
1. The motivations or remaining challenges are not so clear or what kinds of issues or difficulties are this task that is facing. Please give more details and discussion about the key problems solved in this paper, which is largely different from existing works.
2. A deep literature review should be given, particularly advanced and SOTA deep learning. Therefore, the reviewer suggests discussing some currently SOTA works by analyzing the following papers in the revised manuscript, e.g., Multi-feature Fusion: Graph Neural Network and CNN Combining.
3. How about the computational complexity?
4. The compared methods are not sufficient. Some SOTA compared methods should be involved.
5. 6. Some more future directions should be pointed out in the conclusion.
Reviewer 2 Report
I congratulate the authors for the research carried out.
In order to improve the manuscript I allow myself to make some considerations.
I identify a general problem in the paper, the sections indicated that the manuscript must have are not followed. The results and discussion sections should be separated and the authors mix them. I advise separating them into clearly differentiated sections.
The contributions of the research should not appear in the introduction (that is what the conclusions section is for) and should be removed from this section.
Abbreviations must be identified the first time they appear in the text. For example, in line 89 ML and DL appear without having previously described what they mean. Another example is RNN or LSTM.
Likewise, the meaning of the abbreviations used in the tables should be described at the foot of the table.
The last paragraph of section 2 should be integrated into the discussion.
It would be interesting if the authors included in the discussion the limitations
Finally, indicate that the conclusion should include only what is related to the results found. Future lines of work should be integrated into the discussion.
Reviewer 3 Report
The article describes a method that labels apnea events relying on oximetry raw data only. The contribution of this work is a new transformer-based neural network architecture with a novel learnable positional encoding, using for the labeling process only a pulse oximeter, which is a small handheld device that clips onto a finger or earlobe. Furthermore, it has superior performance over existing solutions.
- ML and DL (abbreviated form) appears in the text for the first time on line 89, and the complete form appears only on line 101. The authors must move the Machine Learning (ML) and Deep Learning (DL) from line 101 to line 89;
- Authors must inform the quantity of NaN values they eliminated;
- The Base-model must be represented with the aid of a flowchart;
- The 3.2 Base-model section has three subsections starting with number 1 (1.2.1, 1.2.2, and 1.2.3). It is necessary to correct these errors.
- Despite having placed the reference, the authors must explain how the characteristics in DL are learned during training, without human intervention, because this is difficult to understand for the reader who is not an expert in ML;
- The phrase "We apply Adam as an optimizer" needs a reference;
- On line 369, 1e-5 needs to be written using superscript on the exponent;
- The authors must explain why short-length segments are less likely to be affected by noise and outliers. And why these segments make the model more interpretable and easier to understand than long sequences.
Reviewer 4 Report
Comments for the manuscript sensors-2587839 “Adoption of Transformers Neural Network to Improve the Diagnostic Performance of Oximetry for Obstructive Sleep Apnea”
Comments
This study investigates the adoption of transformers neural network to improve the diagnostic performance of Oximetry for obstructive sleep apnea. I think the paper fits well the scope of the journal and addresses an important subject. However, a number of revisions are required before the paper can be considered for publication. There are some weak points that have to be strengthened. Below please find more specific comments:
*Abstract: The abstract could be expanded a bit. I particular, I suggest adding a couple of sentences highlighting the contributions of this work to the state of the art and the major potential implications from the conducted research.
*Keywords: The keywords seem to be acceptable. Some of them could be more concise though.
*The introduction section seems to be reasonable and provides the necessary background.
*The literature review seems kind of short. Please double check for the most recent and relevant studies published over the last 2-3 years.
*Please make sure that all the adopted assumptions are supported by the relevant references. This will help justifying the adoption of these assumptions.
*The presentation of the proposed methodology could be more detailed. I suggest including the flowchart with the main algorithmic steps and discuss every stem of the algorithm in more detail.
*Please provide more details regarding the input data used throughout this study. More supporting references would be helpful to justify the data selection.
*The manuscript contains quite a lot of figures and tables. Please double check and try to provide a more detailed description of these figures and tables where appropriate to make sure that the future readers will have a reasonable understanding of what these figures and tables represent.
*Future research: The authors primarily used neural networks within the solution framework for the studied decision problem. In the future research, the authors should explore more advanced optimization algorithms for this decision problem. The authors should create a general discussion regarding the importance of advanced optimization algorithms (e.g., new types of hybrid heuristics and metaheuristics, adaptive algorithms, self-adaptive algorithms, island algorithms, polyploid algorithms, hyperheuristics, etc.) for challenging decision problems. There are many different domains where advanced optimization algorithms have been applied as solution approaches, such as online learning, scheduling, transportation, medicine, data classification, and others (not just the decision problem addressed in this study). The authors should create a discussion that highlights the effectiveness of advanced optimization algorithms in the aforementioned domains and their potential applications for the decision problem addressed in this study. The neural networks adopted within the solution framework can be compared with these advanced optimization algorithms as a part of the future research. This discussion should be supported by the relevant references, including but not limited to the following:
An online-learning-based evolutionary many-objective algorithm. Information Sciences 2020, 509, pp.1-21.
An Adaptive Polyploid Memetic Algorithm for scheduling trucks at a cross-docking terminal. Information Sciences 2021, 565, pp.390-421.
A study of ant-based pheromone spaces for generation constructive hyper-heuristics. Swarm and Evolutionary Computation 2022, 72, p.101095.
Reviewer 5 Report
This paper presents the application of an innovative deep-learning architecture for the classification of sleep apnoea from non-invasive SpO2 recordings. The work is well structured and organized and the results are promising given the comprehensive literature review reported in the paper. The reviewer greatly appreciates that the authors have reported the detail of the performance achieved by previous studies in tackling the same task.
The Reviewer, therefore, recommends proceeding with the publication of the paper after responding to some minor revisions that I list below.
- When mentioning wearable devices for monitoring vital parameters, the authors may consider citing the work
Vicente-Samper, José María, et al. "An ML-Based Approach to Reconstruct Heart Rate from PPG in Presence of Motion Artifacts." Biosensors 13.7 (2023): 718.
- In lines 90-91 it is stated that "The use of DL to identify OSA using SpO2 has several benefits. It is non-intrusive, cost-effective, and convenient." The reviewer does not understand whether the subject is the DL algorithm or the SpO2 measurement. The authors should rephrase to clarify better.
- Authors should check that they have referred to acronyms such as DL for the first time in full and always use only the contracted form in the following.
- The results reported in Table 2 and Fig 3 should also report standard deviations. Since the authors used k-fold validation methods, they should have the data to represent not only the mean value obtained on the 5 folds but also the dispersion of the calculated metrics. This would add rigor and completeness to the presentation of the results.
Round 2
Reviewer 1 Report
No more comments!
Reviewer 4 Report
My comments have been adequately addressed.